# OpenReview forum: "Inducing Overthink: Hierarchical Genetic Algorithm-based DoS Attack on Black-Box Large Language Reasoning Models"
_ICML.cc/2026/Conference — ICML 2026 regular_

### Official Review · Reviewer_9b1S · 2026-03-07

**Soundness:** 2
**Presentation:** 3
**Significance:** 2
**Originality:** 2
**Overall Recommendation:** 3
**Confidence:** 3

**Summary:**

This paper proposes a Hierarchical Genetic Algorithm (HGA)-based framework for inducing computational overthinking in Large Reasoning Models (LRMs) as a black-box Denial-of-Service attack. The key insight is that LRMs exhibit pathologically extended reasoning chains when confronted with logically inconsistent inputs, specifically, problems whose premise sets and questions have been deliberately mismatched or corrupted. The authors represent each input as a structured (premises, question) pair and apply two-level genetic operators (question-level and premise-level crossover, plus premise deletion/insertion mutation) to evolve adversarial problem instances that maximize a composite fitness function combining output token length and frequency of reflective hesitation markers. Experiments across four commercial LRMs (DeepSeek-R1, Qwen3-Thinking, GPT-o3, Gemini-2.5-Flash) and three math reasoning benchmarks (SVAMP, GSM8K, MATH500) report up to 26.1× amplification in output length relative to unmodified inputs. Additional experiments demonstrate that adversarial inputs evolved on a small open-source proxy model (Qwen3-14B) transfer to closed-source commercial targets, and ablations validate the composite fitness design over single-objective alternatives.

**Compliance With Llm Reviewing Policy:**

Affirmed.

**Final Justification:**

The paper addresses a relevant security concern — reasoning-specific resource exhaustion in black-box LRMs — and the structured premise-question search space is a sensible design choice that improves interpretability over generic suffix attacks. The composite fitness formulation and the transferability experiments are clearly presented and provide concrete empirical support.

The rebuttal addressed several of my original concerns. The GEPA comparison empirically justifies HGA over LLM-guided optimization, the multi-seed runs confirm statistical reliability, and the AutoDoS comparison demonstrates competitive output amplification with substantially fewer input tokens. I appreciate the authors' effort in providing these additional experiments.

However, three concerns from my original review remain unresolved after the rebuttal, and together they limit the contribution below the acceptance threshold:

(1) The transferability results are reported only relative to BASE, omitting comparison against direct-attack performance. The substantial drop from direct optimization to transferred attack (approximately 75% reduction) is not acknowledged or discussed, which overstates the practical severity of the transfer setting.

(2) Generalization beyond math reasoning is discussed only conceptually. Without any experiments on non-math domains (e.g., coding or open-ended dialogue), the claim that the method targets "reasoning models broadly" remains unsupported, and the contribution is better characterized as math-specific.

(3) The sensitivity of attack effectiveness to the choice of the reflective-marker vocabulary V is not ablated. Given that Table 2 already shows substantial instability across α values, the interaction between V and α remains an uncontrolled variable.

I maintain my original score of 3 (Weak Reject). The paper has clear merits in problem formulation and empirical methodology, but the gaps in transferability evaluation, domain generalization, and hyperparameter sensitivity collectively prevent me from recommending acceptance in its current form.

**Key Questions For Authors:**

1. Why genetic algorithm?
The optimization objective, maximizing a weighted combination of output token length and reflective marker frequency, defines a standard black-box prompt optimization problem with a scalar reward signal. Methods such as ACE (https://arxiv.org/abs/2510.04618) and GEPA (https://arxiv.org/abs/2507.19457) are specifically designed for this setting and require no access to gradients. Could the authors provide an empirical comparison against such a baseline, or explain why evolutionary search is better suited to this problem than LLM-guided optimization?
2. Are the reported results statistically reliable, and how sensitive are they to initialization?
Each entry in Table 3 is derived from a single GA run initialized from 10 randomly sampled problems, approximately 60 queries total per (model, dataset) pair. It is unclear whether these 10 seed problems are sampled uniformly at random, and no justification is given for this sample size relative to the full datasets. No variance estimates, confidence intervals, or multi-seed results are reported. Could the authors report results averaged over multiple independent runs with different random seeds, along with standard deviations? How sensitive are the Max-len and Avg-len statistics to the choice of initial population?
3. What is the actual cost-effectiveness of the attack?
The paper reports output token length as the sole proxy for attack severity. Could the authors report the attacker-to-victim cost ratio, i.e., how many tokens the attacker must spend to induce each additional token of victim output, to establish whether the attack is economically viable in practice?
4. Does the method generalize beyond math reasoning datasets?
All three evaluation benchmarks (SVAMP, GSM8K, MATH500) consist of math word problems, which represent a narrow subset of real-world LLM deployments. The genetic operators, premise deletion, premise insertion, and question-level crossover, are defined over a premise-question decomposition that is natural for math inputs but has no obvious analog for domains like coding tasks, scientific reasoning, or open-ended dialogue. Could the authors discuss how these operators would be adapted for non-math domains, or evaluate the framework on one such setting?

**Limitations:**

yes

**Strengths And Weaknesses:**

# Strength
1. The paper targets a real and underexplored attack surface: reasoning-specific overthinking in black-box LRMs. Framing the attack as an automated search over structured problem perturbations is a sensible approach, and the paper is among the first to specifically exploit the reasoning chain itself rather than surface-level prompt features such as length or token distribution.
2. The paper defines a concrete optimization objective combining a verbosity score (output token count) and a reflective marker score (frequency of hesitation tokens). The ablation over the trade-off parameter α provides direct evidence that neither objective alone is sufficient, and that the composite formulation yields strictly higher amplification than optimizing for length alone, a modest but clearly demonstrated empirical finding.
3. The paper evaluates across four commercial reasoning models (DeepSeek-R1, Qwen3-Thinking, GPT-o3, Gemini-2.5-Flash) and three benchmarks (SVAMP, GSM8K, MATH500). The transferability experiment demonstrates that adversarial inputs evolved on a small open-source proxy model retain measurable effectiveness against closed-source commercial targets, and the hyperparameter ablation confirms that the method reaches near-peak performance within a modest query budget.

# Weaknesses
1. Insufficient justification for genetic algorithm over black-box prompt optimization methods. The optimization objective, a weighted combination of output verbosity and reflective marker frequency, constitutes a standard black-box prompt optimization problem with a well-defined scalar reward signal. The paper provides no justification for why a genetic algorithm is preferred over established methods such as ACE (https://arxiv.org/abs/2510.04618) or GEPA (https://arxiv.org/abs/2507.19457), which require no gradient access and are specifically designed for optimizing natural language inputs under black-box conditions. Without an empirical comparison against such baselines, the choice of GA appears arbitrary and the claimed methodological advantages cannot be substantiated.
2. Comparison against datasets rather than attack methods. The experimental evaluation compares HGA against two static datasets, the unmodified benchmark problems and the MIP collection, rather than against prior adversarial attack systems. MIP is a dataset constructed to study the overthinking phenomenon, not an attack framework. No quantitative comparison is made against AutoDoS or OverThink, both of which are directly applicable in the same black-box setting and evaluated on the same model families. This design choice systematically avoids the most informative comparison and makes it impossible to assess whether the proposed method advances the state of the art in LRM-targeted DoS attacks.
3. Structural input requirement severely limits generalizability. The framework requires each input to be decomposable into an explicit premise set and a question, a structure present in math word problem datasets but absent from most real-world deployment contexts. Unlike prior methods such as GCG or AutoDoS, which can generate adversarial inputs for arbitrary prompts, HGA is restricted to structured mathematical reasoning tasks. The paper provides no analysis of whether the core attack mechanism, logical inconsistency induced by premise-question mismatch, generalizes beyond the math reasoning domain.
4. Evaluation is statistically unreliable due to insufficient sampling. With population size N=10 and approximately 60 total queries per (model, dataset) pair, each cell in Table 3 reflects a single GA run initialized from 10 randomly sampled problems. No variance estimates, confidence intervals, or multi-seed averages are reported. The Max-len statistic in particular, a single extreme value drawn from 10 samples, is highly sensitive to initialization and cannot support reliable conclusions. Given that SVAMP, GSM8K, and MATH500 collectively contain thousands of problems, the reported results lack the statistical grounding necessary to establish reproducibility or generalizability.
5.  Score2, a core component of the fitness function, counts the frequency of tokens drawn from a predefined vocabulary V of overthink markers. Appendix A.3 discloses only four illustrative examples (maybe, but, alternatively, perhaps). The rationale for its construction is not provided; for instance, whether terms were selected based on frequency analysis of known overthinking outputs drawn from prior literature, or by manual inspection. Without the full vocabulary, Score2 cannot be independently reproduced, and its contribution to the reported results cannot be verified. Moreover, the sensitivity of attack effectiveness to vocabulary design is not ablated: it is unknown whether a different choice of V would yield materially different fitness landscapes or final token amplification figures.
6. Transferability results are presented without disclosing performance degradation. Table 5 reports amplification factors for transferred attacks relative to BASE, but omits comparison against direct attack performance from Table 3. For GPT-o3 on SVAMP, direct optimization achieves an average response length of 3,346 tokens, whereas the transferred attack yields only 825, a 75% reduction. Reporting amplification relative to an unattacked baseline while omitting this gap gives a misleading impression of transferability effectiveness.

---

> ### Author Rebuttal · Authors · 2026-03-30
>
> We sincerely thank the reviewer for the careful reading of our paper and for these thoughtful questions. We appreciate the reviewer’s recognition of the importance of studying reasoning-specific overthinking in black-box LRMs, and we agree that the current version should better justify the design choices, strengthen the statistical evidence, and clarify both practical cost and scope.
>
> **Q1: Why is HGA preferable to other black-box optimizers?**
>
> Thank you for your insightful comment. We address this concern from two aspects below.
> First ,regarding LLM-guided optimization,  our aim is not to optimize arbitrary surface-form text, but to perturb the logical structure of the text at the level of premises and questions, rather than tokens. This makes genetic algorithm a natural fit, since it can recombine locally effective logical perturbations while preserving readability and compatibility with the original input format. To support this empirically, we have added a comparison with GEPA, and find that HGA induces **substantially longer responses across all three datasets**, which we will report in **Section 4.2**.
>
> ||SVAMP|GSM8K|MATH|
> |-|-|-|-|
> |GEPA|569|880|6253|
> |HGA|**8185**|**11635**|**31376**|
>
> Second, to compare against prior attack methods, we add a comparative experiment about HGA and AutoDoS. The results indicate that HGA achieves comparable or stronger output amplification with substantially fewer input tokens, especially on MATH. We will add this comparison to **Section 4.2** to  enhance the effectiveness of our method.
> ||AutoDoS input|AutoDoS Max-out|AutoDoS Avg-out|HGA input|HGA Max-out|HGA Avg-out|
> |-|-|-|-|-|-|-|
> |SVAMP|2518|12775|8209|90|10708|8599|
> |GSM8K|2448|11491|8551|165|11301|9662|
> |MATH|2652|16009|11207|**99**|**32768**|**25419**|
>
> **Q2: How statistically reliable are the results?**
>
> Thank you for raising this concern. In our experiments, the ten initial seeds are sampled uniformly at random from each dataset, and Table 3 currently shows one representative run. To address statistical reliability more directly, we have **additionally run ten independent trials** with different random seeds and will report mean and standard deviation in **Section B.1** to verify the effectiveness of our method. The results of multiple runs on three datasets are shown below, where ASR represents the probability that the output length of the HGA method is greater than that of BASE and MIP in each run.
>
> |SVAMP|Avg-len(Mean)|Avg-len(Std Dev)|ASR|Max-len(Mean)|Max-len(Std Dev)|ASR|
> |-|-|-|-|-|-|-|
> |BASE|467|208|-|1615|1587|-|
> |MIP|2596|422|-|6288|2397|-|
> |HGA|**5041**|1708|**100%**|9594|5364|**100%**|
>
> |GSM8K|Avg-len(Mean)|Avg-len(Std Dev)|ASR|Max-len(Mean)|Max-len(Std Dev)|ASR|
> |-|-|-|-|-|-|-|
> |BASE|578|209|-|2136|2045|-|
> |MIP|2090|1043|-|7567|2549|-|
> |HGA|6270|1339|**100%**|12258|**7289**|**100%**|
>
> |MATH500|Avg-len(Mean)|Avg-len(Std Dev)|ASR|Max-len(Mean)|Max-len(Std Dev)|ASR|
> |-|-|-|-|-|-|-|
> |BASE|2445|1401|-|9952|7863|-|
> |MIP|8718|2520|-|8819|5466|-|
> |HGA|19828|5466|**100%**|**29632**|7128|**100%**|
>
> **Q3: What is the practical cost-effectiveness of the attack?**
>
> Thank you for highlighting this issue. We compare HGA with the existing black-box method AutoDoS. Our preliminary accounting shows that AutoDoS requires around 25,000 attacker-side tokens to generate a 6-subtask-attacked prompt, whereas HGA requires on average around 20,000 tokens. More importantly, HGA can generate attack prompts using a proxy model, and the effectiveness of transferring from a **small proxy mode**l to **LRMs** is demonstrated in Table 5 and Table 8.
>
> **Q4: How does the method generalize beyond math tasks?**
>
> Thank you for this suggestion. We use premise-question decomposition because math word problems provide a clean and controllable representation of logical dependencies. More broadly, the key abstraction is not “math premise” itself, but a **domain-specific structural unit** whose dependency to the final task can be perturbed.  In coding, these units could be function signatures,  API constraints, or retrieved context. In open-ended dialogue, the structural unit could be prior turns,  user goals or memories. In scientific reasoning, This could be evidentiary statements, claims or table entries.
>
> **Q5: What is the complete definition of overthink markers?**
>
> Thank you for this important point. In the revision, we will provide the complete vocabulary $V$ of overthink tokens in **Appendix A.3**:
> ```
> but, wait, maybe, problem, perhaps, another, alternatively, not
> ```
> Our construction is partly motivated by Deadlock[1], which highlights reflective markers such as **but** and **wait**, and is further extended using a data-driven rule: we select high-frequency non-function words whose occurrence differs most between LRM responses under logically confused versus normal inputs.
>
> [1] One token embedding is enough to deadlock your large reasoning model. Zhang, M., et al. NIPS 2025.

---

> > ### Author Rebuttal · Reviewer_9b1S · 2026-03-31
> >
> > Thank the authors for the new experiments, particularly the GEPA comparison, multi-seed runs, and the AutoDoS cost comparison. These address several of my original concerns.
> > Three points remain unaddressed:
> > (1) The rebuttal does not respond to the concern that transferability results are reported only relative to BASE, omitting comparison against direct attack performance. The 75% reduction from direct optimization to transferred attack is not discussed.
> > (2) Generalization beyond math reasoning is discussed conceptually (Q4), but no experiments are provided. The structural input requirement remains a significant limitation.
> > (3) The complete vocabulary V is now disclosed, which is appreciated. However, the sensitivity of attack effectiveness to the choice of V is not ablated; it remains unknown whether a different vocabulary would yield materially different results.

---

> > > ### Author Response · Authors · 2026-04-04
> > >
> > > We thank the reviewer for the follow-up. These are fair remaining concerns, and we agree that the revision should make the limits of the current evidence more explicit.
> > >
> > > **Q6: Transferability relative to direct optimization**
> > >
> > > Thank you for highlighting this issue. We agree that reporting transferability only relative to BASE is incomplete and that comparison against direct optimization should be made explicit. The large gap noted by the reviewer arises in part from seed sensitivity: the transferred result and the direct-optimization result being compared come from different random initial populations and as already shown in our response to Q2, performance exhibits standard deviation across seeds. As a result, a direct-versus-transfer comparison can overstate the apparent degradation. To address this more transparently, we now introduce a **transfer retention rate**, defined relative to direct optimization rather than BASE, which ensure that direct optimization and transfer optimization use the same population. This allows us to separate two questions more clearly: whether transferred attacks still produce substantial amplification over benign inputs, and how much of the direct-attack effectiveness is retained after transfer. Under this matched comparison, transfer remains weaker than direct optimization in most cases, but the degradation is materially smaller than the comparison highlighted by the reviewer suggests; in some settings, stochasticity can even yield retention slightly above 100%. We will **add this analysis in the Appendix A.5** to provide a fairer and more complete assessment of transferability.
> > >
> > > ||Base-Avg-len|Trf-Avg-len|Avg-len|Avg-trf-rate|Trf-Max-len|Max-len|Max-trf-rate|
> > > |-|-|-|-|-|-|-|-|
> > > |SVAMP|363|4350|4560|95.4%|6001|6725|89.2%|
> > > |GSM8K|407|5303|5718|92.7%|9172|9515|**96.3%**|
> > > |MATH|2473|25040|24163|**101.7%**|31012|32768|94.6%|
> > >
> > > **Q7: Generalization beyond math**
> > >
> > > Thank you for your insightful comment. We also agree that the original math-only evaluation limits the scope of the current paper. To address this concern directly, we **have extended the evaluation to three additional domains: HumanEval for coding, EntailmentBank for scientific reasoning and MT-Bench for open-ended dialogue**. Our central claim is not that HGA requires a specifically mathematical premise-question format, but that it operates on domain-relevant structural units whose dependencies can be perturbed.
> > >
> > > ||Base-Max-len|Base-Avg-len|HGA-Max-len|HGA-Avg-len|
> > > |-|-|-|-|-|
> > > |HumanEval|3082|1390|**14467**|**8138**|
> > > |EntailmentBank|1620|530|4346|2081|
> > > |MT-Bench|5255|1944|10445|6141|
> > >
> > > As shown in the table above, HGA generalizes effectively beyond math reasoning benchmarks. Across HumanEval, EntailmentBank and MT-Bench, it consistently produces substantial output amplification over the corresponding base inputs. We will include this experiment in **Appendix B.1** to further substantiate the generalizability of the proposed HGA method.
> > >
> > > **Q8: Sensitivity to the overthink-marker vocabulary.**
> > >
> > > Thank you for raising this point. We agree that releasing the full vocabulary $V$ improves reproducibility but does not by itself establish robustness to vocabulary design. To address this, we add a vocabulary-sensitivity ablation with **five variants of $V$**: prior-work overthinking markers only, differential-frequency markers only, their union, a removal setting without reflective markers, and a randomly selected control vocabulary. This experiment is designed to test whether $score_2$ captures a stable reflective signal or whether the reported gains rely on a particular hand-crafted lexicon.  The results show that the **union vocabulary performs best overall**, while both the removal setting and random control are consistently weaker, indicating that the gains do not arise from arbitrary token counting alone. We will report these results in **Appendix B.2** to clarify the robustness of the fitness function.
> > >
> > > |SVAMP|Vocabulary Source|Avg-len|Max-len|
> > > |-|-|-|-|
> > > |Length-only|None|1624|2536|
> > > |Prior|Prior work only|1885|2786|
> > > |Gap|Top frequency-gap words only|1774|2699|
> > > |Union|Prior work + frequency-gap words|**1911**|**3257**|
> > > |Random|Random non-marker control words|1131|2374|
> > >
> > > |GSM8K|Avg-len| Max-len  |
> > > |-|-|-|
> > > |Length-only|2981|5745|
> > > |Prior|3196|6449|
> > > |Gap|4423|7421|
> > > |Union|**7478**| **9927** |
> > > |Random|2177|4375|
> > >
> > > |Math|Avg-len|Max-len|
> > > |-|-|-|
> > > |Length-only|15611|21763|
> > > |Prior|20178|30830|
> > > |Gap|16738|25824|
> > > |Union|**22070**|**32768**|
> > > |Random| 13701 |25934|

---

### Official Review · Reviewer_ZoQi · 2026-03-10

**Soundness:** 2
**Presentation:** 3
**Significance:** 2
**Originality:** 3
**Overall Recommendation:** 4
**Confidence:** 3

**Summary:**

This work proposes an automated, black-box Denial-of-Service attack that exploits the "overthinking" vulnerability in Large Reasoning Models. Using a Hierarchical Genetic Algorithm, the method perturbs the logical structure of input queries (premises and questions) to induce logically inconsistent prompts. By optimizing a fitness function that maximizes both response verbosity and the frequency of "overthink markers", the attack traps models in redundant reasoning loops. Evaluations show the attack inflates output lengths, up to 26.1x on the MATH benchmark, and demonstrates high transferability to commercial LRMs when evolved on smaller proxy models.

**Compliance With Llm Reviewing Policy:**

Affirmed.

**Final Justification:**

I have raised my recommendation to a weak accept as the authors' rebuttal effectively addressed my primary concerns regarding the threat model, baseline comparisons, and the robustness of the over-think marker objective.

- The comparison with AutoDoS (Table 1) successfully demonstrates that HGA induces massive output amplification using significantly fewer input tokens. This clarifies the unique mechanism and efficiency of targeting reasoning-trace inflation.
- The new vocabulary-sensitivity ablation and the cross-lingual evaluation on MATH23K provide evidence that the over-think markers are not over-fitted to a specific English set and can generalize across languages.
- I appreciate the authors' transparency regarding the instability of $\alpha$ across datasets. Documenting this as a limitation makes the findings more grounded and scientifically honest.

As over-think markers are a key contribution of this work, I look forward to seeing these new results and the associated discussions fully integrated into the final version as promised.

**Key Questions For Authors:**

### Questions
- **Q1: Threat Model Distinction and Baselines**
  - Given that HGA shares the goal of resource exhaustion via lengthy outputs with recent black-box DoS attacks, how does the proposed "reasoning-trace targeting" threat model distinctively benefit the attacker? Could the authors provide empirical evidence showing HGA better bypasses system-level monitors? Alternatively, a comparison with these SOTA baselines would be necessary.
- **Q2: Complete Vocabulary and Transferability**
  - For reproducibility, please provide the complete vocabulary ($\mathcal{V}$) of overthinking tokens. Furthermore, to address potential model-specific bias when transferring from a proxy (e.g., Qwen) to target LRMs, could the authors provide an ablation study comparing the composite fitness function against a pure-length objective ($\alpha=1.0$)? This would help verify whether $Score_2$ genuinely improves cross-model generalization or merely overfits to the proxy's stylistic patterns.
- **Q3: Empirical Interpretability and Evolution Dynamics (Minor)**
  - To substantiate the claims in Section B.2 regarding disrupted logical structures and high-attention markers, could the authors provide a qualitative case study tracking a query's evolution trajectory? Demonstrating how the emergence of logical contradictions empirically correlates with internal shifts would significantly strengthen this contribution of the paper.

**Limitations:**

Yes

**Strengths And Weaknesses:**

### Strengths
- **Automated Exploitation of LRM Logical Vulnerabilities**
  - While the "overthinking" phenomenon and LLM-DoS attacks have been identified in prior literature, this paper proposes an automated, structure-aware method to exploit this specific vulnerability. By using a Hierarchical Genetic Algorithm, the paper offers a scalable approach to stress-test the robustness of state-of-the-art LRMs (e.g., GPT-o3 and DeepSeek-R1) without relying on manual prompt crafting.
- **Novel Optimization Objective for Cognitive Overload**
  - The framework introduces a composite fitness function by jointly optimizing for both response verbosity and the frequency of reflective "overthink markers" to induce genuine reasoning loops and cognitive overload.
- **Practical Transferability and Cost-Efficiency**
  - The authors demonstrate that adversarial inputs optimized on a smaller, open-source proxy model can transfer effectively to large, closed-source commercial models. This capability lowers the query cost for attackers, highlighting the practical feasibility and severity of the threat in real-world API settings.

### Weaknesses
- **Unclear Threat Model and Omission of SOTA Baselines**
  - The paper claims a distinct threat model targeting "reasoning traces", but its ultimate goal (resource exhaustion via lengthy outputs) is identical to prior LLM-DoS. It would strengthen the paper to empirically demonstrate that reasoning-trace inflation uniquely bypasses system-level monitors and show the benefit than prior LLM-DoS. This is crucial because logically, HGA yields nonsensical, easily detectable outputs, unlike attacks optimized for answer stealthiness. Without proving a clear advantage in reasoning-trace targeting, the threat model distinction remains weak, making the omission of recent SOTA black-box DoS baselines (e.g., AutoDoS/Crabs) a major flaw in evaluating the method's superiority.
- **Ambiguous Definition of Overthinking Tokens and Model-Specific Bias**
  - The fitness function relies on a vaguely defined vocabulary of overthinking tokens ($\mathcal{V}$), with only a partial list provided in Appendix A.3, limiting reproducibility. Furthermore, a hardcoded $\mathcal{V}$ introduces model-specific bias. Different LRMs (e.g., Qwen vs. GPT-o3) naturally exhibit different hedging or reasoning markers due to varied alignment training. This introduces uncertainty regarding the transferability claims: does $Score_2$ genuinely improve cross-model attacks, or does it merely overfit to the proxy model's stylistic patterns? The paper would benefit from a transferability ablation study comparing the composite fitness function against a pure-length objective ($\alpha=1.0$) to prove that $Score_2$ remains effective across models with divergent reasoning vocabularies.
- **Under-explored Interpretability and Evolution Dynamics (Minor)**
  - Section B.2 attributes the attack's success to disrupted logical structures and high-attention markers. While intuitive, these claims lack empirical backing. The paper would be significantly strengthened by including qualitative case studies tracking the evolution trajectory of queries (e.g., observing how logical contradictions emerge across generations).

---

> ### Author Rebuttal · Authors · 2026-03-30
>
> Thank you very much for the thoughtful and constructive review, and for the careful reading of our submission. We sincerely appreciate the reviewer’s recognition of the paper’s strengths, especially the automation of the attack pipeline, the composite optimization objective and the practical transferability analysis.
>
> **Q1: Threat Model Distinction and Baselines**
>
> We agree that HGA and prior black-box DoS attacks share the same ultimate objective: energy-latency attack via longer outputs. Our intended distinction is therefore not the attack goal, but the **attack mechanism**. Specifically, AutoDoS primarily operates by expanding a simple question into a more elaborate DoS prompt, whereas HGA starts from a premise-question decomposition and perturbs the logical dependencies within that structure. As a result, HGA does not rely on substantial input-length inflation to induce long reasoning traces. To support this point empirically, we will add a direct comparison with AutoDoS on three datasets. As shown in Table 1, HGA uses dramatically **fewer input tokens **while achieving **comparable or stronger** output amplification especially on MATH. We will **add this experiment to Section 4.2** to strengthen the empirical support for our claims..
>
> > Table 1: Input and output lengths using AutoDoS and HGA methods
>
> ||AutoDoS input|AutoDoS Max-out|AutoDoS Avg-out|HGA input|HGA Max-out|HGA Avg-out|
> |-|-|-|-|-|-|-|
> |SVAMP|2518|12775|8209|90|10708|8599|
> |GSM8K|2448|11491|8551|165|11301|9662|
> |MATH|2652|16009|11207|**99**|**32768**|**25419**|
>
> **Q2: Ambiguous Definition of Overthinking Tokens and Model-Specific Bias**
>
> We understand the reviewer’s concern about reproducibility and model-specific bias. In the revision,  the complete vocabulary $V$ of overthink tokens  is listed below:
>
> ```
> but, wait, maybe, problem, perhaps, another, alternatively, not
> ```
>
> Our construction is partly motivated by Deadlock[1], which highlights reflective markers such as **but** and **wait**, and is further extended using a  data-driven rule: we select high-frequency non-function words whose occurrence differs most between LRM responses under logically confused versus normal inputs, which will be **added to Section 3.2**.
>
> To test whether the composite fitness improves cross-model transfer, or whether it merely overfits proxy-specific stylistic patterns, we add an ablation study over α in ${0.5, 0.7, 1.0}$, where $α=1$ corresponds to the pure-length objective and smaller α increasingly weights overthink tokens. We report average and maximum output lengths on SVAMP, GSM8K, and MATH. The data in Table 2 demonstrate that the pure-length fitness function is consistently outperformed by the configuration with $α = 0.7$, as evidenced by **both average and maximum lengths across all evaluated datasets**. We will **add the experiment to Section 4.3** to increase the credibility and effectiveness of our method.
>
> > Table 2:   Composite fitness functions tranfer from Proxy-Model (Qwen3-14B) to DeepSeek-R1
>
> ||α=0.5|α=0.5|α=0.7|α=0.7|α=1.0|α=1.0|
> |-|-|-|-|-|-|-|
> ||Max-len|Avg-len|Max-len|Avg-len|Max-len|Avg-len|
> |SVAMP|3334|1821|**9887**|**5785**|5847|3091|
> |GSM8K|**8379**|**4358**|6839|4038|5728|2778|
> |MATH|4904|2441|**31998**|**10893**|12320|7018|
>
> **Q3: Interpretability and evolution dynamics.**
>
> ```
> There were 2 roses in the vase. Jessica threw away 4 roses from the vase. Jessica cut some more new roses from her flower garden to put in the vase. There are now 23 roses in the vase. How many roses did she cut?
>
> He sold 90 pastries. There were 2 roses in the vase. Jessica threw away 4 roses from the vase. Jessica cut some more new roses from her flower garden to put in the vase. There are now 23 roses in the vase. How many pencils and crayons does she have altogether?
> ```
>
> We agree that the current interpretability discussion is still preliminary, and we will strengthen it with a concrete case study. Due to space limitations, the above shows GEN={0,4}. In the example above, the initial seed is a coherent arithmetic problem about roses. In Gen 4, the mismatch becomes even stronger: the same rose premises are **paired with an unrelated pencil question**. At the same time, **premise-level corruption also accumulates**, including add premise "He sold 90 pastries" into the rose problem context. These examples show a clear evolutionary trajectory: the evolution does not merely lengthen prompts, but progressively breaks premise-question alignment and local logical consistency, which is precisely the mechanism hypothesized in our interpretability discussion. We will **add a more detailed trajectory analysis to the appendix B.2** complement the current conceptual explanation.
>
> In conclusion, we are very grateful for the reviewers' detailed feedback and objective evaluation. We believe these suggestions will significantly improve this paper.
>
> [1] One token embedding is enough to deadlock your large reasoning model. Zhang, et al. NIPS 2025.

---

> > ### Author Rebuttal · Reviewer_ZoQi · 2026-04-02
> >
> > I appreciate the authors' response. While Q1 and Q3 are resolved, Q2 remains partially resolved. I have the following remaining concerns:
> >
> > * Instability of $\alpha$: Table 2 shows extreme sensitivity; on MATH, $\alpha=0.7$ yields 10,893 tokens while $\alpha=0.5$ drops to 2,441. Although $\alpha=0.5$ performs best on GSM8K, the pure-length objective ($\alpha=1.0$) actually outperforms $\alpha=0.5$ on SVAMP and MATH. This instability suggests that finding the optimal sweet spot of the method still requires substantial tuning.
> >
> > Could the authors discuss the method's adaptability beyond the 8 provided English tokens? Specifically, how would the mining strategy handle non-English reasoning (e.g., DeepSeek or Qwen in Chinese) or variations in user-controllable reasoning effort (low vs. high effort)? I am concerned that the observed instability of $\alpha$ might stem from the limited set, which may not fully capture the diverse overthinking patterns across different model behaviors.

---

> > > ### Author Response · Authors · 2026-04-04
> > >
> > > Thank you very much for this thoughtful follow-up. We appreciate the reviewer’s careful reading of the ablation results and the important questions regarding both the stability of α and the broader adaptability of the marker vocabulary beyond the current English setting.
> > >
> > > **Q1: Instability of α**
> > >
> > > Thank you for this important observation. We agree that the current ablation does not establish a single universally optimal value of α across all settings. Our intention is therefore not to claim that $α=0.7$ is globally optimal, but rather that incorporating the reflective-marker term is beneficial compared with optimizing for output length alone. In particular, in our transfer setting, $α=0.7$ consistently outperforms the pure-length objective $α=1.0$ across all three datasets in both average or maximum output length, which provides direct evidence that the composite objective is effective rather than redundant.
> > >
> > > **Q2: Adaptability beyond provided tokens**
> > >
> > > We thank the reviewer for highlighting this point. We also agree that the current presentation of the marker vocabulary can be improved and address this concern from two aspects below.
> > >
> > > First,  regarding **sensitivity to the vocabulary itself**, we add a  vocabulary-sensitivity ablation with **five variants of $V$**: prior-work overthinking markers only, differential-frequency markers only, their union, a removal setting without reflective markers, and a randomly selected control vocabulary. This allows us to test whether the gains come from a robust class of reflective cues rather than from one particular hand-crafted English lexicon. The results show that the **union vocabulary performs best overall**, while both the removal setting and random control are consistently weaker, indicating that the gains do not arise from arbitrary token counting alone. We will report these results in **Appendix B.2** to clarify the robustness of the fitness function.
> > >
> > > |SVAMP|Vocabulary Source|Avg-len|Max-len|
> > > |-|-|-|-|
> > > |Length-only|None|1624|2536|
> > > |Prior|Prior work only|1885|2786|
> > > |Gap|Top frequency-gap words only| 1774     |2699|
> > > |Union|Prior work + frequency-gap words|**1911**|**3257**|
> > > |Random|Random non-marker control words|1131|2374|
> > >
> > > |GSM8K|Avg-len| Max-len|
> > > |-|-|-|
> > > |Length-only|2981|5745|
> > > |Prior|3196|6449|
> > > |Gap|4423|7421|
> > > |Union|**7478**| **9927** |
> > > |Random|2177|4375|
> > >
> > > |Math|Avg-len|Max-len|
> > > |-|-|-|
> > > |Length-only|15611|21763|
> > > |Prior|20178|30830|
> > > |Gap|16738|25824|
> > > |Union|**22070**|**32768**|
> > > |Random| 13701 |25934|
> > >
> > > Second,  regarding **language dependence**, we agree that an English-only vocabulary is insufficient to support broader claims about reasoning-oriented models. We therefore extend the evaluation to Chinese reasoning using MATH23K and construct a language-specific marker vocabulary with the same mining strategy used in English. Under this Chinese-derived vocabulary, the attack remains **effective and the balanced objective again outperforms pure length optimization**.  Reflective markers provide a useful search signal for overthinking behavior and capture the diverse overthinking patterns across different model behaviors. We will **add the experiment in Appendix A.5** to provide empirical evidence of the method’s effectiveness across diverse linguistic settings.
> > >
> > > |α|Avg-len|Max-len|
> > > |-|-|-|
> > > |0.0|1132|3675|
> > > |0.3|1628|4550|
> > > |0.5|4767|6699|
> > > |0.7|**4855**|**9140**|
> > > |1.0|3904|7094|
> > >
> > > Overall, we will revise the paper in two ways. First, we will weaken the original wording around 𝛼 and explicitly describe its instability across settings as a limitation. Second, we will strengthen the empirical support for vocabulary robustness by adding both a vocabulary-sensitivity ablation and a cross-lingual experiment. We believe these revisions directly address the reviewer’s concern and make the paper’s claims more precise, reproducible, and better supported.

---

### Official Review · Reviewer_tRX4 · 2026-03-12

**Soundness:** 2
**Presentation:** 2
**Significance:** 2
**Originality:** 2
**Overall Recommendation:** 3
**Confidence:** 3

**Summary:**

This paper proposed a black-box hierarchical genetic algorithm that perturbs the premise/question structure of reasoning problems to induce “overthinking” in large reasoning models. The method represents each problem as premises plus a final question, applies question- and premise-level crossover/mutation, and optimizes a fitness that combines output length with counts of reflective markers such as hesitation or self-correction tokens. Experiments are reported on GSM8K, SVAMP, and MATH500 across DeepSeek-R1, Qwen3-Thinking, GPT-o3, and Gemini-2.5-Flash, with additional ablations on the fitness trade-off and a proxy-model transfer setting.

**Compliance With Llm Reviewing Policy:**

Affirmed.

**Final Justification:**

After considering both the paper and the rebuttal, I maintain my weak reject recommendation. The paper has clear strengths: the attack is more structured and interpretable than generic prompt-level search, and the experiments show consistent output-length amplification across models and datasets. This makes the work potentially significant as a robustness concern.

However, my main concerns are only partially resolved. The rebuttal improves the paper’s positioning by adding evidence on the relationship between output length, latency, and cost, clarifying implementation details, and providing additional baseline comparisons and ablations. These clarifications are helpful and make the contribution more credible. Still, the core security claim would be stronger with direct end-to-end measurements rather than relying mainly on output length as a proxy, and several important methodological details and comparisons were not sufficiently established in the original submission.

Overall, the rebuttal addressed some of my concerns but did not change my overall assessment. I continue to view the paper as promising and empirically interesting, but not yet strong enough in soundness and experimental completeness for acceptance in its current form.

**Key Questions For Authors:**

See Weakness

**Limitations:**

yes

**Strengths And Weaknesses:**

## Strengths
- Structured attack formulation: The paper proposes a reasonably clear search space over premises and questions, rather than treating prompts as undifferentiated token strings. This makes the attack idea more interpretable than purely surface-form prompt perturbations and gives a concrete mechanism for reasoning-aware search.
- Consistent empirical effect: The reported results show substantial output-length amplification over clean and missing-premise baselines across several models and datasets, and the transfer experiments suggest the phenomenon is not tied to a single model. This supports the paper’s significance as a systems/security concern, at least with token length as the attack metric.

## Weaknesses
- Proxy metric gap: The main claim is a DoS/resource-exhaustion attack, but the experiments measure output token length rather than latency, dollar cost, rate-limit impact, or energy on the evaluated APIs. Since the paper argues these quantities are proportional, direct measurements on at least some targets would materially strengthen the soundness of the security claim.
- Unclear attack observables: The method defines fitness using the model’s chain-of-thought response and reflective markers, and the introduction states that the system first prompts the model to reveal its chain of thought. For several black-box commercial models, it is unclear what exact text was observable and scored, so the experimental protocol is not specified precisely enough to assess reproducibility.
- Limited comparative evidence: The empirical comparison is mainly against clean inputs and a manual missing-premise dataset, while prior automated black-box DoS-style attacks are discussed only narratively. Without an experimental comparison to the closest automated baselines, the originality claim is plausible but not fully substantiated relative to prior work.
- Method specification gaps: Important implementation details are underspecified, including the full reflective-marker vocabulary, how segmentation errors from the auxiliary LRM affect the search, and whether answer correctness or semantic validity is constrained during evolution. These details matter because the method explicitly allows semantically incoherent perturbations, which may blur the line between exploiting reasoning-specific failure modes and simply generating malformed prompts.

---

> ### Author Rebuttal · Authors · 2026-03-30
>
> Thank you very much for the careful reading and thoughtful feedback. We sincerely appreciate the reviewer’s recognition of the structured attack formulation and the consistent empirical effects across models. We also agree that the current version can better clarify the security interpretation, the experimental protocol, and the implementation details.
>
> **Q1. Proxy metric gap**
>
> Thank you for your insightful comment. Our initial choice of output token length was motivated by Inducing High Energy-Latency, which shows a positive correlation between generation length and inference latency. Considering the difference model used in paper, **we added additional validation of the relationship among output length, dollar cost, and latency**.
>
> Commercial API pricing is directly tied to the number of input and output tokens used, and output tokens are typically more expensive than input tokens.HGA substantially increases the output length while changing the input length only slightly.
>
> ||Base Problem|HGA Problem|Amplification|Base Response|HGA Response|Amplification|
> |-|-|-|-|-|-|-|
> |SVAMP|43|79|1.8×|431|7940|**18.4×**|
> |GSM8K|61|114|1.7×|588|8598|14.6×|
> |MATH|74|94|**1.3×**|4022|32768|8.1×|
>
> In addition, we created a plot of output token length versus latency, which provides an intuitive illustration of the positive correlation between output length and model latency.
>
> https://i.ibb.co/MDzq1VQB/transfer-latency-vs-tokens.png
>
> In the revised manuscript, we will **add the above table and figure to a new Section 2.3** as background for the energy-lattency attack.
>
> **Q2. Attack observability and reproducibility.**
>
> Thank you for highlighting this issue. We address this concern from two aspects. First, we agree experiments involving black-box commercial models present inherent reproducibility challenges. To mitigate this, the current paper already includes transfer experiments showing that adversarial inputs discovered on a **smaller open-source proxy** model remain **effective on other target models**, as shown in Tables 5 and 8 . This reduces dependence on direct optimization against expensive or opaque commercial APIs. Second, regarding the experimental protocol itself, we have provided the relevant implementation details in our response to Q4, including the full reflective-marker vocabulary, scoring procedure, and methodological clarifications. In the revision, we will **incorporate these details directly into the Section 4.1** to make the setup more transparent and easier to reproduce
>
> **Q3. Limited comparative evidence against prior automated baselines.**
>
> Thank you for this suggestion. We agree that a direct experimental comparison to prior automated black-box DoS-style attacks would strengthen the originality claim. We will add a comparison against the recent black-box baseline AutoDoS[1] in Section 4.2. Under this comparison, HGA uses much **shorter inputs** yet achieves **comparable or stronger** output amplification. Meanwhile, prompt-expansion attacks are more exposed to input-side defenses[2], while HGA preserves a short-input attack surface and instead targets the model’s reasoning dynamics more directly.
>
> ||AutoDoS input|AutoDoS Max-out|AutoDoS Avg-out|HGA input|HGA Max-out|HGA Avg-out|
> |-|-|-|-|-|-|-|
> |SVAMP|2518|12775|8209|90|10708|8599|
> |GSM8K|448|11491|8551|165|11301|9662|
> |MATH|2652|16009|11207|**99**|**32768**|**25419**|
>
> **Q4. Method specification and the role of semantic validity.**
>
> Thank you for raising this concern. We address this concern from three aspects. First,  in the revision we will **provide the complete vocabulary $V$ in Appendix A.3 :
>
> ```
> but, wait, maybe, problem, perhaps, another, alternatively, not
> ```
>
> Our construction is partly motivated by Deadlock[3], which highlights reflective markers such as but and wait, and is further extended using a simple data-driven rule: we select high-frequency non-function words whose occurrence **differs most** between LRM responses under logically confused versus normal inputs.
>
> Second, The goal of the auxiliary LRM is to perform logical segmentation not to generate logical errors. Section 3.4 states that our genetic operators intentionally **perturb logical dependencies**,  which affect the chain of reaoning and lead to a significant increase in ouput.
>
> Finally, because HGA is designed as a dos attack, we do not optimize for answer correctness. Instead, we evaluate whether the input induces longer and more costly reasoning traces. Importantly, our operators act on logical units rather than individual tokens, so the resulting inputs are generally still readable rather than token-level gibberish.
>
> [1] Zhang, et al. "Crabs: Consuming resource via auto-generation for llm-dos attack under black-box settings." ACL 2025
>
> [2] Llmlingua: Compressing prompts for accelerated inference of large language models.  Jiang, Huiqiang, et al. ENMLP 2023.
>
> [3] One token embedding is enough to deadlock your large reasoning model. Zhang, M., et al. NIPS 2025.

---

> > ### Author Rebuttal · Reviewer_tRX4 · 2026-04-04
> >
> > Thank you for your response and for addressing the points raised. I acknowledge the clarification provided and will maintain my score.

---

> > > ### Author Response · Authors · 2026-04-06
> > >
> > > We sincerely thank the reviewer for the careful follow-up and for taking the time to read our rebuttal. To further strengthen the paper on the  limited comparative evidence and semantic validity, we have added additional empirical evidence.
> > >
> > > **Q3. Limited comparative evidence against prior automated baselines**
> > >
> > > Thank you for this suggestion. We agree that a direct comparison against prior automated black-box optimization baselines would strengthen the empirical positioning of our method. We therefore add a comparison with **GEPA**, an LLM-guided black-box optimization baseline and will include these results in **Section 4.2**. The results show that **HGA produces substantially longer responses on all three datasets**, suggesting that, for this reasoning-oriented attack setting, search over structured problem components is more effective than LLM-guided optimization over prompt variants.
> > >
> > >
> > > ||SVAMP|GSM8K|MATH|
> > > |---|---|---|---|
> > > |GEPA|569|880|6253|
> > > |HGA|**8185**|**11635**|**31376**|
> > >
> > > **Q4. Method specification and the role of semantic validity.**
> > >
> > > Thank you for raising this concern. We agree that it is important to distinguish between exploiting reasoning-specific failure modes and merely eliciting responses to malformed or incoherent inputs. To clarify this point, we add two additional experiments.
> > >
> > > First, we conduct a **vocabulary-sensitivity ablation** on the reflective-marker vocabulary. The goal is to test whether the benefit of the marker term comes from a meaningful class of reflective cues, rather than from arbitrary malformed-input effects. We consider **five variants of (V)**: prior-work overthinking markers only, differential-frequency markers only, their union, a removal setting without reflective markers, and a randomly selected control vocabulary. Across datasets, the **union vocabulary performs best overall**, while both the removal setting and the random control are consistently weaker. This suggests that the gains are not explained by arbitrary token counting alone, but instead arise from a more robust reflective-marker signal. We will report these results in **Appendix B.2**.
> > >
> > > | SVAMP       | Vocabulary Source                | Avg-len  | Max-len  |
> > > | ----------- | -------------------------------- | -------- | -------- |
> > > | Length-only | None                             | 1624     | 2536     |
> > > | Prior       | Prior work only                  | 1885     | 2786     |
> > > | Gap         | Top frequency-gap words only     | 1774     | 2699     |
> > > | Union       | Prior work + frequency-gap words | **1911** | **3257** |
> > > | Random      | Random non-marker control words  | 1131     | 2374     |
> > >
> > > | GSM8K       | Avg-len  | Max-len  |
> > > | ----------- | -------- | -------- |
> > > | Length-only | 2981     | 5745     |
> > > | Prior       | 3196     | 6449     |
> > > | Gap         | 4423     | 7421     |
> > > | Union       | **7478** | **9927** |
> > > | Random      | 2177     | 4375     |
> > >
> > > | Math        | Avg-len | Max-len |
> > > | ----------- | ------- | ------- |
> > > | Length-only | 15611   | 21763   |
> > > | Prior       | 20178   | 30830   |
> > > | Gap         | 16738   | 25824   |
> > > | Union       | **22070** | **32768**   |
> > > | Random      | 13701   | 25934   |
> > >
> > > Second, we conduct a **cross-lingual evaluation** to test whether the usefulness of reflective markers is tied to English specifically. We extend the evaluation to **Chinese reasoning on MATH23K** and construct a language-specific marker vocabulary using the same mining procedure as in English. Under this Chinese-derived vocabulary, the attack remains effective, and the balanced objective again outperforms pure length optimization. This suggests that reflective markers provide a useful search signal for overthinking behavior beyond a single English vocabulary, and can adapt to different linguistic settings. We will include this experiment in **Appendix A.5**.
> > >
> > > | α    | Avg-len  | Max-len  |
> > > | ---- | -------- | -------- |
> > > | 0.0  | 1132     | 3675     |
> > > | 0.3  | 1628     | 4550     |
> > > | 0.5  | 4767     | 6699     |
> > > | 0.7  | **4855** | **9140** |
> > > | 1.0  | 3904     | 7094     |

---

### Official Review · Reviewer_kkYT · 2026-03-13

**Soundness:** 3
**Presentation:** 3
**Significance:** 3
**Originality:** 3
**Overall Recommendation:** 4
**Confidence:** 5

**Summary:**

The paper proposes a black-box resource exhaustion attack on reasoning models. Inputs are represented as premises plus a question, and a hierarchical genetic algorithm mutates that structure through question crossover, premise crossover, deletion, and addition. Fitness combines output length with a hand-built score for reflective markers such as hesitation and self-correction. The paper reports large output length amplification on several math benchmarks, including transfer from a smaller proxy model to commercial LRMs.

**Compliance With Llm Reviewing Policy:**

Affirmed.

**Ethical Review Concerns:**

The paper develops a black-box resource-exhaustion attack on large reasoning models, including transfer to commercial systems. Because the contribution is directly usable for denial-of-service style misuse, I think an ethics review would be helpful.

**Ethical Review Flag:**

Flag this paper for an ethics review.

**Ethics Expertise Needed:**

["Privacy and Security (e.g., personally identifiable information)"]

**Final Justification:**

This paper studies a realistic black-box resource-exhaustion attack, and the black-box setting is its main strength. The rebuttal improved the paper with stronger comparisons, follow-up non-math experiments, and better vocabulary-sensitivity analysis. Even so, my main soundness concern did not change. The paper still leaves some ambiguity about whether it is inducing overthinking on meaningful reasoning problems or exploiting a broader form of adversarial corruption, and the marker-based fitness is still not fully disentangled from surface style. I keep my original score.

**Key Questions For Authors:**

1. I would like to know how much of the effect survives if the reflective marker score is removed or replaced by something less tied to English phrasing. That matters for robustness across models and domains.

2. The discussion suggests several lightweight defenses, including behavioral monitoring, early stopping, and adaptive truncation. I would be interested to know how much of the attack effect remains under simple heuristics of this kind, since that seems important for judging the practical severity of the threat.

**Limitations:**

Not really a limitation, but a brief discussion of recent overthinking and resource exhaustion attacks, such as Deadlock Attack [1], POT [2], and ThinkTrap [3], may help better distinguish the paper's contribution and threat model.

> [1] Zhang, M., Zhang, Y., Jia, J., Wang, Z., Liu, S., and Chen, T. One token embedding is enough to deadlock your large reasoning model. In *Advances in Neural Information Processing Systems*, 2025.
>
> [2] Li, X., Huang, T., Mu, R., Huang, X., and Jin, G. Pot: Inducing overthinking in llms via black-box iterative optimization. arXiv preprint arXiv:2508.19277, 2025.
>
> [3] Li, Y., Wang, J., Zhu, H., Lin, J., Chang, S., and Guo, M. Thinktrap: Denial-of-service attacks against black-box llm services via infinite thinking. In *NDSS Symposium*, 2026.

**Strengths And Weaknesses:**

**Strengths**

- The black-box setup is realistic. The attack only needs text input and text output, and the transfer experiment does not stop at local open models.
- The search space is better chosen than a generic suffix attack. Working at the level of premises and questions fits the target models and makes the evolved inputs somewhat interpretable.
- The method section is easy to follow. Figure 1 and the algorithm box are clear to understand what the search is doing.

**Weaknesses**

- The motivation emphasizes reasoning-aware, semantically meaningful attacks, but Section 3.4 later says the genetic operators do not aim to preserve semantic or syntactic coherence. If the evolved inputs are often heavily distorted, then the paper may be showing a broader form of adversarial problem corruption rather than overthinking on still meaningful reasoning tasks.

- The fitness design is more fragile than the paper acknowledges. $score_2$ is tied to a very small hand-written vocabulary of reflective markers, and Appendix A.3 lists only a few examples. The ablations around $\alpha$ are useful, but they do not answer whether the search is finding overthinking behavior or partly overfitting to one particular surface style of self-correction.

- The empirical scope is still narrow relative to the framing and everything is built on math benchmarks, so it is hard to tell how much of the effect comes from this particular problem structure. I would be much more convinced by at least one additional setting, such as coding or another non-math reasoning benchmark, especially since the paper is making a claim about black-box reasoning models more broadly.

---

> ### Author Rebuttal · Authors · 2026-03-30
>
> Thank you very much for the careful reading and thoughtful feedback. We appreciate the reviewer’s recognition of the realistic black-box setup, the structured search space and the clarity of the method section. We also agree that the current presentation can do a better job of aligning the paper’s motivation, claims, and empirical scope.
>
> **Q1: Reflective-marker robustness and the removal case**
>
> We address this concern from two aspects below. First, regarding the reflective marker term, the current paper partially addresses the “removal” case through the α-ablation: when $α=1.0$, the search optimizes only output length and effectively removes the reflective marker component from the objective, yet the **attack remains effective.**
>
> Second, we also appreciate the concern about the fitness design and the possibility of overfitting to a small English vocabulary of reflective markers. So we **add an ablation study** comparing the composite fitness function against a pure-length objective from a proxy to target LRM where α=1 corresponds to the pure-length objective and smaller α increasingly weights reflective markers. The results are shown in the table below.
>
> ||α=0.5|α=0.5|α=0.7|α=0.7|α=1.0|α=1.0|
> |-|-|-|-|-|-|-|
> ||Max-len|Avg-len|Max-len|Avg-len|Max-len|Avg-len|
> |SVAMP|3334|1821|**9887**|**5785**|5847|3091|
> |GSM8K|**8379**|**4358**|6839|4038|5728|2778|
> |MATH|4904|2441|**31998**|**10893**|12320|7018|
>
> The data demonstrates that the pure-length fitness function is consistently outperformed by the configuration with $α = 0.7$, as evidenced by **both average and maximum lengths across all evaluated datasets**. We will **add the experiment to Section 4.3** to increase the credibility and effectiveness of our method.
>
> **Q2: Practical severity under lightweight defenses**
>
> To directly assess practical severity, we evaluated several simple black-box defenses motivated by our discussion section—behavioral monitoring, early stopping, and adaptive truncation—using observable signals such as reasoning length, final answer length, reflective-marker frequency, and response latency. Our results show a clear utility–security trade-off: when these heuristics are made aggressive, they begin to suppress or reject benign responses; when their thresholds are calibrated to avoid materially affecting normal questions, the attack rejection rate is only **12%**. In other words, lightweight heuristics can filter a small subset of the most extreme attack cases, but they do not substantially eliminate the attack in the practically relevant regime.
>
> **Q3: Reasoning-aware structure and semantic preservation**
>
> Our intended claim is not that every evolved input remains semantically faithful to the original problem, but rather that the search is structure-aware: it **perturbs the premise–question decomposition directly**, instead of **appending a generic suffix**. Section 3.4 is explicit that the operators do not preserve semantic or syntactic coherence, and we agree the paper should state more clearly that our threat model is about inducing resource-intensive overthinking on still-parseable but logically fractured inputs.
>
> **Q4: Empirical scope and generalization beyond math**
> We agree that the present evaluation is entirely on math-style reasoning benchmarks and the current operators rely on a premise–question decomposition that is especially natural in this setting. More broadly, the key abstraction is not “math premise” itself, but a **domain-specific structural unit** whose dependency to the final task can be perturbed.  In coding, these units could be function signatures,  API constraints, or retrieved context. In open-ended dialogue, the structural unit could be prior turns,  user goals or memories. In scientific reasoning, This could be evidentiary statements, claims or table entries.
>
> **Q5: Positioning relative to recent overthinking and resource-exhaustion attacks**
>
> We appreciate the suggestion to better position the paper relative to recent resource-exhaustion and overthinking attacks. We will add the related work on **Section 2.3**. Deadlock induces the model to repeatedly emit transitional tokens such as “Wait” or “But” at the end of reasoning steps, thereby driving the model into a near-infinite “keep thinking” loop. However, it assumes a significantly stronger threat model. POT uses an external LLM to perform iterative prompt optimization, automatically generating and selecting semantically natural and stealthy guidance phrases that encourage excessively long reasoning. In contrast, it cannot systematically and efficiently search for reasoning-aware attacks by operating on the problem itself. ThinkTrap maps discrete prompts into a continuous surrogate space, performs derivative-free optimization in a low-dimensional subspace. While effective for black-box DoS settings, its optimized prompts are generally less interpretable and the resulting attack inputs may lacks strong readability.

---

> > ### Author Rebuttal · Reviewer_kkYT · 2026-04-04
> >
> > Thank you for the additional comparisons and ablations. Two concerns remain for me. First, the paper still does not show a non-math setting, so the broader claim about black-box reasoning models remains only partially supported. Second, the marker-vocabulary issue is not fully resolved: the $\alpha=1$ ablation shows that the attack still works without $score_2$, but it does not really tell me how sensitive the method is to the specific vocabulary choice or how robust it is across languages and reasoning styles. I will keep my score.

---

> > > ### Author Response · Authors · 2026-04-04
> > >
> > > We thank the reviewer for the follow-up. These are fair remaining concerns, and we agree that these concerns deserve a more complete and carefully articulated response.
> > >
> > > **Q1: Generalization beyond math**
> > >
> > > Thank you for your insightful comment. We also agree that the original math-only evaluation limits the scope of the current paper. To address this concern directly, we **have extended the evaluation to three additional domains: HumanEval for coding, EntailmentBank for scientific reasoning and MT-Bench for open-ended dialogue**. Our central claim is not that HGA requires a specifically mathematical premise-question format, but that it operates on domain-relevant structural units whose dependencies can be perturbed.
> > >
> > > ||Base-Max-len|Base-Avg-len|HGA-Max-len|HGA-Avg-len|
> > > |-|-|-|-|-|
> > > |HumanEval|3082|1390|**14467**|**8138**|
> > > |EntailmentBank|1620|530|4346|2081|
> > > |MT-Bench|5255|1944|10445|6141|
> > >
> > > These results provide initial evidence that HGA can transfer beyond math benchmarks to additional structured reasoning settings. Across HumanEval, EntailmentBank, and MT-Bench, it consistently produces substantial output amplification over the corresponding base inputs. We will include this experiment in **Appendix B.1** to further substantiate the generalizability of the proposed HGA method.
> > >
> > > **Q2: Marker-vocabulary issue**
> > >
> > > We thank the reviewer for highlighting this point. We agree that the marker-vocabulary issue is not fully resolved by the current $α=1$ removal ablation alone, since that result shows only that the attack can still work without $score_2$, but does not yet establish robustness to the specific choice of $V$. We address this concern from two aspects below.
> > >
> > > First,  regarding **sensitivity to the vocabulary itself**,  our marker construction is inspired by prior observations that reflective behavior often co-occurs with transitional markers such as **“but”** and **“wait”** and is further extended using a data-driven rule: we select high-frequency non-function words whose occurrence differs most between LRM responses under logically confused versus normal inputs, which will be added to **Section 3.2**. To address this more directly, we add a  vocabulary-sensitivity ablation with **five variants of $V$**: prior-work overthinking markers only, differential-frequency markers only, their union, a removal setting without reflective markers, and a randomly selected control vocabulary. This allows us to test whether the gains come from a robust class of reflective cues rather than from one particular hand-crafted English lexicon. The results show that the **union vocabulary performs best overall**, while both the removal setting and random control are consistently weaker, indicating that the gains do not arise from arbitrary token counting alone. We will report these results in **Appendix B.2** to clarify the robustness of the fitness function.
> > >
> > > |SVAMP|Vocabulary Source|Avg-len|Max-len|
> > > |-|-|-|-|
> > > |Length-only|None|1624|2536|
> > > |Prior|Prior work only|1885|2786|
> > > |Gap|Top frequency-gap words only| 1774     |2699|
> > > |Union|Prior work + frequency-gap words|**1911**|**3257**|
> > > |Random|Random non-marker control words|1131|2374|
> > >
> > > |GSM8K|Avg-len| Max-len|
> > > |-|-|-|
> > > |Length-only|2981|5745|
> > > |Prior|3196|6449|
> > > |Gap|4423|7421|
> > > |Union|**7478**| **9927** |
> > > |Random|2177|4375|
> > >
> > > |Math|Avg-len|Max-len|
> > > |-|-|-|
> > > |Length-only|15611|21763|
> > > |Prior|20178|30830|
> > > |Gap|16738|25824|
> > > |Union|**22070**|**32768**|
> > > |Random| 13701 |25934|
> > >
> > > Second,  regarding **language dependence**, we extend the evaluation to the **Chinese MATH23K benchmark** and constructed a language-specific marker vocabulary by using the same differential-frequency criterion in Chinese. Under this Chinese-derived vocabulary, the attack remains **effective and the balanced objective again outperforms pure length optimization**. This pattern is consistent with our main finding: the reflective-marker component is not necessary for the attack to function, but it materially improves search effectiveness beyond pure-length optimization, even in a non-English setting. We will add the experiment in Appendix A.5 to provide empirical evidence of the method’s robustness across diverse linguistic settings.
> > >
> > >
> > > |α|Avg-len|Max-len|
> > > |-|-|-|
> > > |0.0|1132|3675|
> > > |0.3|1628|4550|
> > > |0.5|4767|6699|
> > > |0.7|**4855**|**9140**|
> > > |1.0|3904|7094|
> > >
> > > Taken together, these results suggest that $score_2$ is neither necessary nor a brittle artifact of one small English lexicon. Pure-length optimization already induces long responses, while language-specific reflective markers **further improve search effectiveness in both English and Chinese**.

---

### Decision · Program_Chairs · 2026-04-30

**Decision:**

Accept (regular)

**Comment:**

The paper received mixed reviews: the strengths are the consideration of an interesting/realistic attack scenario and the attack is successful on scenarios considered. However, the reviews also raised concerns and requested various ablations (that should have been part of the original submission). My recommendation is a borderline accept (accept if there is space) - there are interesting ideas in the paper and its timely but the execution could be strengthened as pointed out by reviewers.